# Sibling Resemblance in Physical Activity Levels: The Peruvian Sibling Study on Growth and Health

**DOI:** 10.3390/ijerph20054210

**Published:** 2023-02-27

**Authors:** Carla Santos, José Maia, Sara Pereira, Olga Vasconcelos, Rui Garganta, J. Timothy Lightfoot, Go Tani, Donald Hedeker, Peter T. Katzmarzyk, Alcibíades Bustamante

**Affiliations:** 1Centre of Research, Education, Innovation and Intervention in Sport (CIFI2D), Faculty of Sport, University of Porto, 4200-450 Porto, Portugal; 2Research Center in Sport, Physical Education, and Exercise and Health (CIDEFES), Faculty of Physical Education and Sports, Lusófona University, 1749-024 Lisboa, Portugal; 3Department of Health and Kinesiology, Texas A&M University, College Station, TX 77845, USA; 4Motor Behavior Laboratory, School of Physical Education and Sports, University of São Paulo, São Paulo 05508-030, Brazil; 5Department of Public Health Sciences, University of Chicago, Chicago, IL 60637, USA; 6Pennington Biomedical Research Center, Baton Rouge, LA 70808, USA; 7School of Physical Education and Sports, National University of Education Enrique Guzmán y Valle, 60637 La Cantuta, Lurigancho-Chosica 15472, Peru

**Keywords:** physical activity, siblings, shared factors, multilevel modelling, Peru

## Abstract

Physical activity is associated with a host of positive health outcomes and is shaped by both genetic and environmental factors. We aim to: (1) estimate sibling resemblance in two physical activity phenotypes [total number of steps∙day^−1^ and minutes for moderate steps per day (min∙day^−1^)]; and (2) investigate the joint associations of individual characteristics and shared natural environment with intra-pair sibling similarities in each phenotype. We sampled 247 biological siblings from 110 nuclear families, aged 6–17 years, from three Peruvian regions. Physical activity was measured using pedometers and body mass index was calculated. In general, non-significant variations in the intraclass correlation coefficients were found after adjustment for individual characteristics and geographical area for both phenotypes. Further, no significant differences were found between the three sib-ship types. Sister-sister pairs tended to take fewer steps than brother-brother (*β* = −2908.75 ± 954.31). Older siblings tended to walk fewer steps (*β* = −81.26 ± 19.83), whereas body mass index was not associated with physical activity. Siblings living at high-altitude and in the Amazon region had higher steps/day (*β* = 2508.92 ± 737.94; *β* = 2213.11 ± 776.63, respectively) compared with their peers living at sea-level. In general, we found no influence of sib-types, body mass index, and/or environment on the two physical activity phenotypes.

## 1. Introduction

Physical activity (PA) has been associated with a variety of positive health outcomes, generating transitional benefits from childhood through adolescence into adulthood [1,2], including decreases in obesity [3], cardiovascular disease, and diabetes [4], and increases in cognitive function as well as academic achievement [5]. Despite its recognized benefits, updated information on the prevalence and trends in PA [6,7] showed that the majority of children and adolescents worldwide are physically inactive, putting their current and future health at risk, and Peru is no exception [8]. For example, a recent global estimate from 146 countries showed that 81% of children and adolescents aged 11–17 years were physically inactive [9], with the prevalence of insufficient physical activity in Peruvian youth increasing from 82.6% in 2001 to 84.7% in 2016. To reverse current trends, it is important to investigate what types of factors can effectively influence daily active play and PA behaviors in childhood and adolescence.

There is considerable variability among children in their level and patterns of PA, and this variability is shaped by a host of genetic [10,11,12] and environmental [13,14] factors. Behaviors such as PA are often influenced by household and family characteristics, as families often share common interests and experiences [15,16,17]. However, different family members also express a degree of autonomy when it comes to lifestyle behaviors, and sometimes variation among related subjects is also remarkable [18].

The study of siblings offers unique insights into their biology and behavior, given their relationships to each other, as well as to other family members. For example, siblings share a substantial fraction of their genes that are transmitted from their parents; in addition, siblings often grow and mature in similar environments including the household, school, and neighborhood contexts. In addition, they also differ in their chronological age, maturity status, sex, body composition, physical fitness, or lifestyle choices [19,20,21].

Genome-wide association studies (GWAS) have provided evidence that variation in PA is associated with polymorphisms in several genes [22,23,24]. However, several reviews of the extant literature do not identify specific genetic factors exclusively responsible for physical activity phenotypes [25]. On the other hand, variation in sibling resemblance depending on the sib-type and the phenotype has also been considered. For example, Pereira, et al. [26], using questionnaire data in Portuguese sibling pairs, showed that after adjustments for several covariates (biological, behavioural, familial, and environmental characteristics), sister-sister pairs demonstrated greater resemblance in their PA (*ρ* = 0.53) than brother-sister (*ρ* = 0.26) or brother-brother pairs (*ρ* = 0.18). In contrast, Jacobi, et al. [27] found no differences in correlations between siblings (all *ρ* = 0.28) when using PA data collected with pedometers.

Since PA is a complex and multifaceted trait, it has also been documented that a significant fraction of its variation can be explained by different environmental exposures throughout the lifespan [28], and this is particularly evident in developing countries like Peru. The distinct living settings of Peruvians have been recognized as a kind of “natural laboratory”, a singular territory that offers an opportunity to assess the impact of geographical variation on PA levels by combining settings on the spectrum of both rural-urban developments as well as lowland-highland scenarios. Peruvians are exposed daily to different natural stressors (e.g., altitude, temperature, pollutants), as well as social and economic inequalities (e.g., access to health care, quality of nutrition, access to public recreational infrastructure) in a unique geographical diversity [29], which can influence intrapair similarities in PA levels. To date, there is only one study focusing on physical fitness phenotypes in Peru, which concluded that both individual characteristics and geographical area of residence were significantly related to the magnitude of sibling resemblance as well as the mean levels of physical fitness [21].

Despite this recognition, to date there is no available evidence regarding variation in PA levels among Peruvian siblings, especially embracing the diversity of the three distinct geographical areas. Hence, using sibling data, as well as a multilevel statistical approach [30], we explored resemblance in PA levels among Peruvian siblings conditioned on the additive effects of their individual characteristics and shared natural environment. Specifically, we intend to: (1) estimate sibling resemblance in two PA phenotypes [total number of steps∙day^−1^ and minutes for moderate steps (min∙day^−1^)]; and (2) investigate the joint associations of individual characteristics (age and body mass index) as well as a shared natural environment.

## 2. Materials and Methods

### 2.1. Design and Participants

Our sample originates from The Peruvian Sibling Study on Growth and Health [31]. This study probes into sibling resemblance in body composition, physical fitness, physical activity, different facets of their motor development as well as gross motor coordination. A total of 247 biological siblings [(147 females and 100 males from 110 nuclear families (67.2% two siblings; 32.8% three siblings)] were selected. All are native to three Peruvian geographical areas located at different altitudes: sea-level (Barranco = 58 m), Amazon region (La Merced and San Ramon = 751 m), and high-altitude (Junín = 4107 m). Only families that had two or three children, aged between 6 and 17 years, with complete PA data were considered in the present paper. Parents or legal guardians provided written informed consent. The project was approved by the Ethics Committee of the School of Physical Education and Sports, National University of Education Enrique Guzmán y Valle, Peru (UNE EGyV). Following their approval, all known siblings were invited to participate in the study.

### 2.2. Measurements and Tests

#### 2.2.1. Anthropometry

Body measurements were made according to standardized protocols [32]. Height was measured with a portable stadiometer (Sanny, Model ES-2060) holding the child′s head in the Frankfurt plane, to the nearest 0.1 cm; weight was measured with a digital scale (Pesacon, Model IP68), with a precision of 0.1 kg. Body mass index (BMI) was calculated using the standard formula: BMI = [weight(kg)/height(m)^2^].

#### 2.2.2. Physical Activity

In order to objectively measure PA, we used pedometers, a body movement sensor that validly and reliably assesses PA among children and youth [33,34]. Pedometers have been used in different populations from different countries [35], and their validity has been studied [36]. Subjects used the Omron Model Walking style II pedometer (Omron Healthcare, Inc., Muko, Japan) over five consecutive days (three weekdays and two weekend days). These pedometers have a multiday memory function that automatically stores the total number of steps∙day ^−1^ (a proxy measure of the total volume of PA), and the walking time, in minutes (min∙day ^−1^), at a moderate or brisk pace in a day (a proxy measure of moderate-to-vigorous PA—this counts the amount of time spent walking at 3.0 METs or more) [37]. Siblings were instructed in the use of the pedometer, learning to remove it only for bathing and before sleeping at night. The devices were attached to the trouser belt (strap) using a clip, leaving the unit perpendicular to the ground. For the present study, only data from sib-ships with complete information from five consecutive days (Wednesday to Sunday) with an average of 12 h∙day ^−1^ of pedometer use were considered.

#### 2.2.3. Shared Environment Characteristics (Natural Environment)

Given the country’s heterogeneity in geographical terms, participants came from the three distinct regions located at different altitudes: sea-level, Amazon region, and high-altitude. Barranco (58 m) was the chosen city at sea-level and this is one of the 43 districts of Lima Province, located on the shore of the Pacific Ocean. The cities of La Merced and San Ramon (751 m) in the Chanchamayo district represented the Amazon region that is the largest in the Peruvian territory and occupies ~60% of its surface. The Junín district (4107 m) was used to represent the high-altitude location on the southern shore of Lake Junín or Chinchaycocha.

### 2.3. Data Quality Control

Data quality control was enhanced by all assessment team members being systematically trained by the lead researchers of the project to: (i) comply with the correct use of technical body measurement procedures; and (ii) instruct parents and children about the pedometer use protocol and persuade them to follow their regular PA routine. Further, IBM-SPSS v26 software was used to facilitate data entry and to cross-check data elements, employing automatic controls to ensure values were not outside known ranges.

### 2.4. Statistical Procedures

Analysis of the data was conducted in a sequential manner. We first performed data cleaning and initial exploratory analyses to identify outliers and check for normality of distributions. In order to normalize the distribution of the phenotype minutes for moderate steps (min∙day ^−1^), a log transformation was applied and the sum of log-scores was computed. Descriptive statistics for all phenotypes i.e., means and standard deviations, were calculated. Differences between geographical residence areas were examined with analysis of variance (ANOVA). Along with the ANOVA, Tukey HSD tests were used for multiple comparisons. SPSS v26 software was used for all analyses, and the Type-I error rate was set at 5%. As sibling data are clustered, and since individuals are nested within their sib-ships (brother-brother BB, sister-sister SS, brother-sister BS), multilevel models were used for statistical analysis [38]. Separates within and between sib-ship variances were first estimated to comply with our first aim. As such, different intraclass correlation coefficients (*ρ*) with corresponding 95% confidence intervals (95% CI) for each PA phenotype were computed. Further, based on the likelihood-ratio test, we compared a model that constrained *ρ* to be equal across sib-ship pairs (Null model) to a model that freely estimated *ρ* across sib-ship pairs (Model 1). The following models were henceforth estimated with the same or different *ρ*, depending on the result attained from the likelihood-ratio test.

For the second aim, the model was expanded (Model 2) to include individual variables such as age and BMI, with *ρ* being re-estimated for each sib-type. Finally, the full model (Model 3) included the geographical area of residence. For model comparison, the likelihood ratio test was used. Given that there are only three regions (sea-level, Amazon region, and high-altitude), as advocated, we did not treat region as a level in the multilevel model [39]. Instead, dummy variables were used to account for differences attributable to region in the fixed part of Model 3, with sea-level as the reference category. Continuous covariates were mean-centered, and sea-level BB pairs served as the reference category. For the multilevel analyses, STATA 14 software was used, with the Type-I error rate set at 5%.

## 3. Results

Table 1 shows descriptive statistics for all study variables. On average, no statistically significant differences (*p* > 0.05) were found among sib-ship pairs from the three geographical areas for chronological age and height. Further, siblings living in the Amazon region are heavier (F = 5.55, *p* < 0.05), have a higher BMI (F = 13.55, *p* < 0.05), and take more steps∙day^−1^ (F = 21.52, *p* < 0.05) compared with their peers from the other regions. On the other hand, sib-ships from the Amazon region spent fewer minutes for moderate steps (min∙day ^−1^) compared to those at sea-level (F = 3.53, *p* < 0.05).

Table 2 provides estimates for the unadjusted and adjusted sibling’s correlations at each PA phenotype. For both phenotypes, Model 1 did not improve model fit relative to the Null model. Thus, there is insufficient evidence to reject the assumption of equal intraclass correlation for the three sib-pairs (BB, SS and BS). From the null model, total number of steps∙day^−1^ intraclass correlation = 0.44 (95%CI = 0.31–0.58), and minutes for moderate steps (min∙day^−1^) intraclass correlation = 0.35 (95%CI = 0.22–0.51). In general, the inclusion of individual characteristics (Model 2), as well as the different geographical areas, did not significantly influence the size of the intraclass correlations in both phenotypes. Additionally, for the minutes of moderate steps phenotype, the last model (Model 3) was not tested since Model 2 was not better than the previous model (Δ = −157.54, *p* = 0.43).

Table 3 shows the multilevel analysis results. Model 3 fit the data significantly better than Model 2 only for total number of steps∙day^−1^. In general, PA averages for BB pairs are *β* = 11,158.63 ± 1001.06; SS pairs tended to take fewer steps compared with BB (*β* = −2908.75 ± 954.31), while non-significant differences were found between BS and BB pairs (*p* > 0.05). Older siblings tended to take fewer steps (*β* = −81.26 ± 19.83, *p* < 0.05), whereas BMI was not statistically significant (*p* > 0.05). Further, siblings living at high-altitude and in the Amazon region tended to take more steps (β = 2508.92 ± 737.94, *p* < 0.05; *β* = 2213.11 ± 776.63, *p* < 0.05, respectively) compared with those living at sea-level.

## 4. Discussion

The present study is innovative in providing in-country PA data for Peru dedicated to siblings living at different altitudes with their specific socioeconomic characteristics, cultural disparities as well as built and natural environments. Our results showed that differences in Peruvian sib-ships resemblance in two PA phenotypes were mainly influenced, apparently, by genetic factors since non-significant differences were found in the intraclass correlation coefficients after adjustments for individual characteristics and geographical area of residence. Further, no significant differences were found between the three sib-ship types.

The available literature has reported varying results. For example, Jacobi, Caille, Borys, Lommez, Couet, Charles and Oppert [27], using French nuclear family data in conjunction with pedometer PA measurements, reported low correlations (*ρ* = 0.28) among siblings for the number of steps per day, although adjustments were only made for sex and age. On the other hand, Maia et al. [40], using the Baecke questionnaire in Portuguese family data, showed differences in a total PA phenotype between sib-types, with BB resembling more than SS and BS. Pereira, Katzmarzyk, Gomes, Souza, Chaves, Santos, Santos, Bustamante, Barreira, Hedeker and Maia [26], also using Portuguese siblings data and the same PA assessment tool, showed that with increasing levels of covariate adjustments, SS pairs showed stronger resemblance than BS and BB pairs. A similar trend was also found in a recent paper analyzing Peruvian sibling resemblances in physical fitness components. Significant differences across sib-types were only observed for waist circumference and handgrip strength, with BB correlations being higher than the SS or the BS correlations, after adjustments for individual characteristics (including age, height, body mass index, and maturity offset) and geographical area of residence [21]. In summary, we believe that correlation discrepancies between studies may be due to different sampling strategies, diverse covariate adjustments, different statistical techniques used to compute correlations, and the phenotypic expression as well as instruments used.

Some previous genetic studies have attempted to identify specific genes that may regulate PA [22,41]. However, this is not a straightforward task, as heritability estimates for PA have ranged from moderate to very high [10]. For example, in a review by de Vilhena e Santos, Katzmarzyk, Seabra and Maia [12], the authors reported genome-wide linkage data with markers near different PA related genes, while Lightfoot [24] indicated that only 2 candidate genes showed consistent associations in the regulation of PA—dopamine receptor 1 (Drd1) and helixloop helix 2 (Nhlh2). Further, recent GWAS indicated a genetic contribution to PA, with Doherty et al. [42] uncovering 14 *loci* for device-measured PA, while Klimentidis, et al. [43] identified multiple variants in habitual PA including CADM2 and APOE. Notwithstanding this progress, results are still unclear, most probably because of specificities in the production of genome maps in genome-wide linkage studies, uses of different methods to estimate PA, or the different ethnic composition of each sample.

Our multilevel model showed that PA averages for BB pairs are 11,159 steps∙day^−1^, which means that they tend to comply, on average, with the guideline recommendations for children and adolescents [44]. Consistent with our sibling data, chronological age has been negatively associated with PA [26,45]. In our study, for each year increase in sibling age, there was an average reduction of 81 steps∙day^−1^, whereas Pereira, Katzmarzyk, Gomes, Souza, Chaves, Santos, Santos, Bustamante, Barreira, Hedeker and Maia [26], based on self-reported PA, similarly revealed a decrease among Portuguese siblings. Using non-sibling data, Duncan, et al. [46] also showed a decline in the number of steps per day with age among New Zealand children and adolescents (15,284 weekday steps and 12,948 weekend steps at 5–6 years of age to 14,801 weekday steps and 10,656 weekend steps at 11–12 years of age). Using accelerometry, Alvis-Chirinos, et al. [47] also reported a decline in moderate-to-vigorous PA with age among Peruvian youth (1.354 min at 6–9 years to 1167 min at 10–13 years).

Our results also indicated dissimilarities in PA among siblings living in the three geographical areas, which potentially reflect the marked regional variations in terms of sociodemographic, economic, and cultural features. For example, in the city of Barranco, children are exposed to several built constraints like compact urban areas, large population centers, and extensive housing developments, with serious consequences in terms of traffic regulation, not to mention increases in public insecurity and environmental problems. Such local constraints can deprive children of playing freely in the community’s streets without parental supervision, as well as restricting access to public recreational and sports services. This may help to explain the likelihood of sibling pairs walking fewer steps compared with their peers from the other regions. In turn, in Chanchamayo and Junín, children tend to take more steps per day, probably because they find plenty of space for leisure and free playing, helping them to develop their abilities, deepen and widen their experiences, acquire further skills, and discover other interests. However, we could not find a published paper that investigated the links between natural environments (sea-level, Amazon region, or high-altitude) and siblings’ PA to make suitable comparisons.

Notwithstanding the importance of the present data, some limitations must be recognized. Firstly, without data indicating otherwise, it is possible that our sample is not representative of the overall Peruvian sibling population. Secondly, given the study design, genetic and environmental influences could not be estimated separately because no twins were involved. Thirdly, we made no adjustments for family socioeconomic status. While limited, this report also has several unique strengths. Firstly, the study involves a relatively large sample of siblings from three unique environmental contexts, although its size may not have sufficient power to detect putative interactions of different sib-types with their varying environments. Secondly, the study covered both childhood and puberty periods, expanding the range of potential influences from biological and environmental factors. Thirdly, the use of standardized and highly reliable objective methods for data collection makes significant contributions to the available literature. Finally, the use of a multilevel analysis model with individual and environmental data allows for approaching their interaction in the development of PA.

## 5. Conclusions

In conclusion, our model-based results revealed that, in general, there are no significant differences in the intraclass correlation coefficients for both PA phenotypes after adjustment for age and BMI as well as the geographical area of residence. Further, non-significant differences were found between the three sib-ship types. SS pairs tended to take fewer steps∙day^−1^ than BB, while non-significant differences were found between BS and BB pairs. Older siblings tended to walk fewer steps∙day^−1^, whereas BMI was not associated with PA. Further, siblings living at high-altitude and in the Amazon region tended to walk more steps∙day^−1^ compared with their peers living at sea-level.

Overall, our results highlight the significant sibling resemblance effects in explaining variance in PA, with genetic factors apparently being the most important legacy to explain dissimilarity, although environmental features must also be considered.

## Figures and Tables

**Table 1 ijerph-20-04210-t001:** Descriptive statistics [means and standard deviations (SD)], F tests and post-hoc comparisons by each geographical region.

	Sea-Level (S) (*n* = 66)	Amazon Region (A) (*n* = 80)	High-Altitude (H) (*n* = 101)	*F*	*Post Hoc* among Regions	
	Mean ± SD					
Age (years)	10.4 ± 2.9	10.6 ± 3.1	10.5 ± 2.9	0.039 ^ns^		
**Anthropometry**						
Height (cm)	138.2 ± 14.8	135.8 ± 15.0	134.0 ± 15.6	1.51 ^ns^		
Weight (kg)	35.4 ± 12.8	37.1 ± 12.6	31.5 ± 10.5	5.55 *	A > H	
BMI (kg/m^2^)	18.0 ± 3.9	19.6 ± 3.9	17.0 ± 2.3	13.55 *	A > S; A > H	
**Physical activity**						
Total number of steps∙day^−1^	8.155 ± 2878	11.170 ± 3931	11.695 ± 3621	21.52 *	S < A; S < H	
Log transformed minutes for moderate steps (min∙day^−1^)	0.82 ± 0.45	0.61 ± 0.48	0.67 ± 0.49	3.53 *	A < S	

* *p* < 0.05; ^ns^ = non-significant.

**Table 2 ijerph-20-04210-t002:** Unadjusted and adjusted sibling’s correlations (*ρ*) and their 95% confidence intervals for both physical activity phenotypes.

		BB (95%CI)	SS (95%CI)	BS (95%CI)	Log Likelihood (LL)	Δ LL(χ^2^) ^¥^	*p*-Value
Physical activity							
Total number of steps∙day^−1^	Null model: equal *ρ*	0.44 (0.31–0.58)	0.44 (0.31–0.58)	0.44 (0.31–0.58)	−2352.09		
	Model 1 (without covariates and different *ρ*)	0.58 (0.26–0.84)	0.58 (0.36–0.77)	0.38 (0.22–0.57)	−2347.73	4.36 (8.72)	0.06
	Model 2 (individual characteristics)	0.46 (0.34–0.60)	0.46 (0.34–0.60)	0.46 (0.34–0.60)	−2343.21	8.88 (17.76)	<0.001
	Model 3 (geographical area of residence)	0.42 (0.29–0.56)	0.42 (0.29–0.56)	0.42 (0.29–0.56)	−2337.39	5.82 (11.64)	<0.001
Minutes for moderate steps (min∙day^−1^) (log transformed)	Null model: equal *ρ*	0.35 (0.22–0.51)	0.35 (0.22–0.51	0.35 (0.22–0.51	−158.91		
	Model 1 (without covariates and different *ρ*)	0.41 (0.11–0.80)	0.22 (0.05–0.61)	0.40 (0.23–0.59)	−158.15	0.76 (1.52)	0.82
	Model 2 (individual characteristics)	0.34 (0.21–0.50)	0.34 (0.21–0.50)	0.34 (0.21–0.50)	−157.54	1.37 (2.75)	0.43

^¥^ Comparison values between Models; BB = brother-brother; SS = sister-sister; BS = brother-sister.

**Table 3 ijerph-20-04210-t003:** Estimates, standard errors (SE), and variance components (σ^2^) for total number of steps∙day^−1^.

Full Model (Model 3)	
**Fixed effects**	Estimate ± SE
Intercept (BB)	11158.63 ± 1001.06 *
SS	−2908.75 ± 954.31 *
BS	−1393.79 ± 806.13 ^ns^
Age	63.56 ± 72.47 ^ns^
Age^2^	−81.26 ± 19.83 *
BMI	−115.43 ± 66.10 ^ns^
High-altitude	2508.92 ± 737.94 *
Amazon region	2213.11 ± 776.63 *
**Variance components (σ^2^)**	σ^2^ ± SE
Between siblings (σ^2^_B_)	
All sibtypes	4638044 ± 1061023
Within siblings (σ^2^_w_)	
All sibtypes	6306225 ± 758801.7

* *p* < 0.05; ^ns^ = non-significant; BB = brother-brother; SS = sister-sister; BS = brother-sister; BMI = body mass index.

## Data Availability

The data are property of the School of Physical Education and Sports, National University of Education Enrique Guzmán y Valle, Peru (UNE EGyV), as is therefore protected from being freely shared.

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
