# Peer review of "Sibling Resemblance in Physical Activity Levels: The Peruvian Sibling Study on Growth and Health"

_ijerph, 2023, doi:10.3390/ijerph20054210_

Round 1

Reviewer 1 Report

IJERPH-2163372 Santos et al

Sibling resemblance in physical activity levels. The Peruvian Sibling Study on Growth and Health

Summary

The authors have studied sibling resemblance in physical activity (PA) in 247 biological siblings from 110 nuclear families aged 6-17 years (100 boys) from the Peruvian Sibling Study. Participants were stratified over three regions: sea-level, Amazon, and high-altitude. PA has been measured by pedometers (Omron Model Walking style II) and study parameters were  i) steps/d at 3.0 METs or more and ii) walking time (min/d) at 3.0 METs or more on 5 consecutive days (W to S). Sib-ships (BB, SS, and BS) multilevel models were used (within- and between-variances and ICCs). Models (rho values) accounted for age, BMI and area of residence and LHRs were calculated with sea-level BB-pairs as reference. In essence we are dealing with a “negative” study: no important variations in adjusted rho’s were observed between the 3 phenotypes and no significant differences between the 3 sib-ship types. Only a few (significant or trend) associations (in secondary analyses) were noticed. High-altitude and Amazon siblings had higher steps/d and SS phenotype had fewer steps/d than the BB phenotype. Older siblings walked fewer steps than younger siblings.

Comments

1.     We are dealing with an interesting observational study. In essence a “negative” study as the answer to the major study questions was “non-significant”. However it is not the major concern of the reviewer. After reading the manuscript the reviewer has a number of questions and comments.

2.     Only a few “significant” associations (side analyses) have been noticed (Table 3). The reviewer even wonders whether it could not be spurious associations. After reading the manuscript it remains unclear whether the authors had a number of a priori hypotheses in mind on the direction of the studied associations. Did they expect that SS pairs had fewer steps/d than BB pairs or that BMI was not associated with PA? It should be clarified. Or was it a pure observational hypothesis-generating study?

3.     Quality control. Table 1. The mean of log transformed min/d at METs of 3.0 or more was <1 corresponding with about 10 000 steps/d also at METs of 3.0 or more. It is a highly unlikely combination. It should be clarified. If the step of a child is about 0.5m it is a distance of about 5 km moderate activity/d, thus more than 30 min/d moderate walking activity. Even for running a log transformed value <1 is unlikely.

4.     Was the study powered to derive warranted conclusions?

5.     The authors incriminated “genetic factors”. Do they have in mind specific genes e.g. some of the candidate genes as obtained from earlier GWAS studies?

6.     The lack of data on SES is really a limitation as SES may determine the children’s opportunities/restrictions related to PA.

Author Response

Dear Ms. Tiffany Yu

International Journal of Environmental Research and Public Health (IJERPH)

Firstly, we would like to thank you for the opportunity to submit a new version of our manuscript. Secondly, we want to acknowledge the generosity of the three reviewers for their critiques/comments/suggestions (in italics) that helped enhance the quality of the new draft. Two of the reviewers were quite favourable towards the paper and had only a couple of minor suggestions. Please find below our answers, especially those responding to the first reviewer who had more substantial questions. Changes in the new manuscript are marked in yellow.

Comments to the Authors

Reviewer 1

1.We are dealing with an interesting observational study. In essence a “negative” study as the answer to the major study questions was “non-significant”. However, it is not the major concern of the reviewer. After reading the manuscript the reviewer has a number of questions and comments.

Authors answer: We thank the reviewer for her/his generous words regarding our paper. Also, we truly appreciate all comments, suggestions and questions hoping that the new draft now meets IJERPH standards for publication.

  1. Only a few “significant” associations (side analyses) have been noticed (Table 3). The reviewer even wonders whether it could not be spurious associations. After reading the manuscript it remains unclear whether the authors had a number of a priori hypotheses in mind on the direction of the studied associations. Did they expect that SS pairs had fewer steps/d than BB pairs or that BMI was not associated with PA? It should be clarified. Or was it a pure observational hypothesis-generating study?

Authors answer: We thank the reviewer for the truly challenging comment and questions. Please find below our answers:

  1. Sibling and twin studies dealing with physical activity, or physical fitness phenotypes, using a top-down approach are mostly observational. As such, they frequently rely on a priori hypotheses which are, in our view, mostly often based on “educated guesses”. We believe that the main reason is that in physical activity epidemiology research there is a lack of a formal theory, or theories, to guide researchers in their quest. For example, the ecological model suggested in 2006 by Sallis et al., a variant of Urie Bonfenbrenner bio-ecological theory (1977, 1979), does not postulate any substantive body of hypotheses to be tested. Further, Sallis et al. model is so complex that it is untestable in mathematical-statistical terms.
  2. Further, the Sallis et al. model (2006) does not have any design approach akin to Genetic Epidemiology classical designs – top-down or bottom-up. This means that Sallis and his research group did not use twins, siblings or nuclear families together with statistical approaches as the ones implemented in Robert Elston S.A.G.E. software, or Mike Neale Mx software.
  3. When we started the Peruvian Sibling Study on Growth and Health we did not know exactly what to expect, and any putative hypotheses would probably fall “off the mark”. For example, if we go back in “history”, the Freedson paper (1991) on familial aggregation in physical activity does not posit any set of hypotheses conditioned by the nature of their clustered data. Further, in the Jacobi et al (2011) study using nuclear families and pedometers, their aims (please see below in italic) did not explicitly state any substantive hypothesis:

The primary objective of this study was to examine familial aggregation in pedometer-assessed ambulatory activity by assessing parent-offspring correlations under daily life conditions. Another objective was to investigate whether correlations would change with increasing age of the offspring.

  1. The reports that used familial data that were, most probably, “highly demanding” on their statistical analyses were the ones generated from the Quebec Family Study using Province and Rao’s SEGPATH software. Given their families structures and using all available correlations that could be estimated, a set of data-driven hypotheses was tested. Yet, authors of these papers always faced the real challenge of a biological and environmental justification for their hypotheses.

In fact, we did not postulate any hypotheses apart from those model-driven by our statistical analyses.

References:

Bronfenbrenner, U. (1977). Toward an experimental ecology of human development. American psychologist, 32(7), 513.

Bronfenbrenner, U. (1979). The Bronfenbrenner, U. (1979). The ecology of human development: Experiments by nature and design. Harvard university press.

Freedson PS, Evenson S. Familial aggregation in physical activity. Res Q Exerc Sport. 1991 Dec;62(4):384-9. doi: 10.1080/02701367.1991.10607538. Erratum in: Res Q Exerc Sport 1992 Dec;63(4):453. PMID: 1780560.

Jacobi D, Caille A, Borys JM, Lommez A, Couet C, Charles MA, Oppert JM; FLVS Study Group. Parent-offspring correlations in pedometer-assessed physical activity. PLoS One. 2011;6(12):e29195. doi: 10.1371/journal.pone.0029195. Epub 2011 Dec 28. PMID: 22216207; PMCID: PMC3247254.

Pereira S, Katzmarzyk PT, Gomes TN, Souza M, Chaves RN, Santos FK, Santos D, Bustamante A, Barreira TV, Hedeker D, Maia JA. Resemblance in physical activity levels: The Portuguese sibling study on growth, fitness, lifestyle, and health. Am J Hum Biol. 2018 Jan;30(1). doi: 10.1002/ajhb.23061. Epub 2017 Sep 19. PMID: 28925585.

Sallis JF, Cervero RB, Ascher W, Henderson KA, Kraft MK, Kerr J. An ecological approach to creating active living communities. Annu Rev Public Health. 2006;27:297-322. doi: 10.1146/annurev.publhealth.27.021405.102100. PMID: 16533119.

  1. Quality control. Table 1. The mean of log transformed min/d at METs of 3.0 or more was <1 corresponding with about 10 000 steps/d also at METs of 3.0 or more. It is a highly unlikely combination. It should be clarified. If the step of a child is about 0.5m it is a distance of about 5 km moderate activity/d, thus more than 30 min/d moderate walking activity. Even for running a log transformed value <1 is unlikely.

Authors answer: We thank the reviewer for this comment. Perhaps the explanation of this variable was not clear in the methods section and we explained it further. Please note that since the phenotype moderate steps (expressed in terms of min∙day-1) was highly skewed, a log transformation was applied to normalize its distribution, and a sum of log-scores was computed. This is now corrected in the new draft.

Please note also that in Tudor-Locke et al (2011) paper they state “in terms of normative data (i.e., expected values) the update international literature indicates that we can expect 1) among children, boys to average 12,000 to 16,000 steps/day and girls to average 10,000 to 13,000 steps/day; and, 2) adolescents to steadily decrease steps/day until approximately 8,000-9,000 steps/day are observed in 18-year olds”. The results we reported in our table 1, averages of steps∙day-1, are below the number suggested by Tudor-Locke et al (2011). Further, the average increases from the sea-level to high altitude were related to children and adolescents’ distinct living conditions, available recreational/sports activities, as well as their daily chores.

Reference:

Tudor-Locke C, Craig CL, Beets MW, Belton S, Cardon GM, Duncan S, Hatano Y, Lubans DR, Olds TS, Raustorp A, Rowe DA, Spence JC, Tanaka S, Blair SN. How many steps/day are enough? for children and adolescents. Int J Behav Nutr Phys Act. 2011 Jul 28;8:78. doi: 10.1186/1479-5868-8-78. PMID: 21798014; PMCID: PMC3166269.

  1. Was the study powered to derive warranted conclusions?

Authors answer: We thank the reviewer for this question. Indeed, no a priori power analysis was done to identify the minimum sample size to comply with our aims. Even if we had this number, we would always face the problem of free participation and parents’ consent. Please note that previously published papers vary in the number of siblings. For example, in Pérusse et al (1979) the total number of subjects was 347, in Simonen et al (2002) was 384, and in Jacobi et al (2011) was 630. But these studies also had parents.  Please note that we added a note on power in our study limitations/strengths.

References:

  1. The authors incriminated “genetic factors”. Do they have in mind specific genes e.g. some of the candidate genes as obtained from earlier GWAS studies?

Authors answer: At this point, the literature does not suggest any particular genetic factors that we could be implicated [(e.g. Lin et al (2017), Aadahl et al (2021), and de Geus (2022)].  We have changed the manuscript to correct the impression that we have specific genes in mind. Additionally, we included some references in the Introduction and Discussion to indicate the importance of genetic legacy.

References:

Aasdahl L, Nilsen TIL, Meisingset I, Nordstoga AL, Evensen KAI, Paulsen J, Mork PJ, Skarpsno ES. Genetic variants related to physical activity or sedentary behaviour: a systematic review. Int J Behav Nutr Phys Act. 2021 Jan 22;18(1):15. doi: 10.1186/s12966-020-01077-5. PMID: 33482856; PMCID: PMC7821484.

Lin X, Eaton CB, Manson JE, Liu S. The Genetics of Physical Activity. Curr Cardiol Rep. 2017 Oct 18;19(12):119. doi: 10.1007/s11886-017-0938-7. PMID: 29046975.

de Geus, EJC. Genetic pathways underlying individual differences in regular physical activity. Exercise and Sport Sciences Reviews 51(1): p2-18, January 2023. | DOI: 10.1249/JES.0000000000000305

  1. The lack of data on SES is really a limitation as SES may determine the children’s opportunities/restrictions related to PA.

Authors answer: Yes, as we are aware that this is a limitation of our study, and we raised this issue in our paper. In any case, please note that the Peruvian children assessed in the three geographical areas attended public schools. Apart from a putative financial condition that may restrict some children and adolescents to participate in sport within the context of private clubs, we think (“educated guess”) that SES may not hinder their daily physical activities. These are mostly done within the school context, their leisure time and daily chores. Please note also that the Peruvian National Institute of Statistics does not have precise information on children’s physical activity participation at the population (from questionnaires or any objective device) nor about their sports participation.

Reviewer 2 Report

I find this research very interesting. A spell check should be done. I would like to emphasize the specificity of this research with regard to the height at which the respondents live, which makes this research even more interesting.

Author Response

Dear Ms. Tiffany Yu

International Journal of Environmental Research and Public Health (IJERPH)

Firstly, we would like to thank you for the opportunity to submit a new version of our manuscript. Secondly, we want to acknowledge the generosity of the three reviewers for their critiques/comments/suggestions (in italics) that helped enhance the quality of the new draft. Two of the reviewers were quite favourable towards the paper and had only a couple of minor suggestions. Please find below our answers, especially those responding to the first reviewer who had more substantial questions. Changes in the new manuscript are marked in yellow.

Comments to the Authors

Reviewer 2

I find this research very interesting. A spell check should be done. I would like to emphasize the specificity of this research with regard to the height at which the respondents live, which makes this research even more interesting.

Authors answer: We gratefully thank the reviewer for her/his considerate words. A general spell check was done as suggested. We have further emphasized the altitude in the manuscript.

Reviewer 3 Report

Article “Sibling Resemblances in Physical Fitness in Three Distinct Regions in Peru: The Peruvian Sibling Study on Growth and Health”  https://link.springer.com/article/10.1007/s10519-022-10099-7#article-info examines health-related physical fitness generally using the same study and methods as the present paper “Sibling resemblance in physical activity levels.

To estimate the sibling resamlance in two physical activity phenotypes and the joint association of individual characteristics and shared natural environment with intra-pair sib-26 ling similarities in each phenotype.

The Peruvian 2 Sibling Study on Growth and Health”. Authors at minimum should convince the audience that their contribution to the literature is significant, should cite their previous article in introduction as well as compare/discuss its findings in the Discussion section.

It addresses a gap in the literature, however, the article looks at the outcome which is very similar to previously published paper by the authors  

https://link.springer.com/article/10.1007/s10519-022-10099-7

This study adds novel finding of non-significant variations in the intraclass correlation coefficients were found after adjustment for individual characteristics and geographical area and no significant differences were found between the three sib-ship types. 

Author Response

Dear Ms. Tiffany Yu

International Journal of Environmental Research and Public Health (IJERPH)

Firstly, we would like to thank you for the opportunity to submit a new version of our manuscript. Secondly, we want to acknowledge the generosity of the three reviewers for their critiques/comments/suggestions (in italics) that helped enhance the quality of the new draft. Two of the reviewers were quite favourable towards the paper and had only a couple of minor suggestions. Please find below our answers, especially those responding to the first reviewer who had more substantial questions. Changes in the new manuscript are marked in yellow.

Comments to the Authors

Reviewer 3

Article “Sibling Resemblances in Physical Fitness in Three Distinct Regions in Peru: The Peruvian Sibling Study on Growth and Health”  https://link.springer.com/article/10.1007/s10519-022-10099-7#article-info examines health-related physical fitness generally using the same study and methods as the present paper “Sibling resemblance in physical activity levels.

To estimate the sibling resemblance in two physical activity phenotypes and the joint association of individual characteristics and shared natural environment with intra-pair sibling similarities in each phenotype.

“The Peruvian Sibling Study on Growth and Health”. Authors at minimum should convince the audience that their contribution to the literature is significant, should cite their previous article in introduction as well as compare/discuss its findings in the Discussion section.

Authors answer: We thank the reviewer for this suggestion. We re-wrote parts of the text to include this information.

It addresses a gap in the literature, however, the article looks at the outcome which is very similar to previously published paper by the authors   https://link.springer.com/article/10.1007/s10519-022-10099-7

This study adds novel finding of non-significant variations in the intraclass correlation coefficients were found after adjustment for individual characteristics and geographical area and no significant differences were found between the three sib-ship types.

Authors answer: We thank the reviewer for these kind comments. Indeed, the previous paper dealt exclusively with physical fitness phenotypes, while this current study addresses physical activity. Please note that we are dealing with two completely different phenotypes, although dealing with the same sibling sample. Based on prior work from Caspersen et al (1982), and reiterated by the report of the Surgeon General (1998),

  • Physical activity is defined as any bodily movement produced by the contraction of skeletal muscle that increases energy expenditure above de basal level. Physical activity can be categorized in various ways, including type, intensity, and purpose. We could add that physical activity should be psychological gratifying and socially relevant.
  • Physical fitness has been defined in many ways. A general accepted approach is to define physical fitness as the ability to carry our daily tasks with vigor and alertness, without undue fatigue, and with ample energy to enjoy leisure-time pursuits and to meet unforeseen emergencies. Physical fitness thus includes cardiorespiratory endurance, skeletal muscle endurance, skeletal muscle strength, skeletal muscle power, speed, flexibility, agility, balance, reaction time, and body composition.

References:

U.S. Department of Health and Human Services (1998). Physical activity and health. A report of the surgeon general. Jones and Bartlett Publishers.

Caspersen C, Powell KE, Christensen GM (1985). Physical activity, exercise, and physical fitness: definitions and distinctions for health-related research. Public Health Reports. 100:126-131.

Round 2

Reviewer 1 Report

The authors have addressed all questions and comments which have been raised by the reviewer and revised their manuscript. However after reading the manuscript the reviewer still does not understand the log transformed minutes for moderate steps. It needs clarification.

A simple calculation. Suppose 10 000 moderate steps/day. Moderate means approx 100 steps/min. Thus approx 100 minutes. Log10(100) = 2. Thus values below 1 as in Table 1 are unlikely. Otherwise the calculations have been done according to other methodologies. Clarify.

Author Response

Dear Ms. Tiffany Yu

International Journal of Environmental Research and Public Health (IJERPH)

Firstly, we would like to thank you for the opportunity to submit a new version of our manuscript. Two of the reviewers were quite favourable towards the paper and had only a couple of minor suggestions. Please find below our answers. Changes in the new manuscript are marked in yellow.

Comments to the Authors

Reviewer 1

The authors have addressed all questions and comments which have been raised by the reviewer and revised their manuscript. However after reading the manuscript the reviewer still does not understand the log transformed minutes for moderate steps. It needs clarification.

A simple calculation. Suppose 10 000 moderate steps/day. Moderate means approx 100 steps/min. Thus approx 100 minutes. Log10(100) = 2. Thus values below 1 as in Table 1 are unlikely. Otherwise the calculations have been done according to other methodologies. Clarify.

Authors answer: Once more we thank the reviewer not only for her/his considerate and generous words but also for the suggestion for further clarification. Please note that,

  • There are a few changes in the manuscript marked in yellow. Hopefully, it will be even more precise regarding the phenotypes we used in the paper.
  • We have now basic descriptive stats (means and standard deviations) for minutes of moderate steps, i.e., we show raw data in minutes as well as their log transformation.
  • The total number of steps also include the number of steps done at moderate intensity.

We hope that now this will clarify the reviewer question.

Reviewer 3 Report

I am thankful for addressing my comments. 

Author Response

Dear Ms. Tiffany Yu

International Journal of Environmental Research and Public Health (IJERPH)

Firstly, we would like to thank you for the opportunity to submit a new version of our manuscript. Two of the reviewers were quite favourable towards the paper and had only a couple of minor suggestions. Please find below our answers. Changes in the new manuscript are marked in yellow.

Comments to the Authors

Reviewer 3

I am thankful for addressing my comments.

Authors answer: We gratefully thank the reviewer for her/his contributions that helped enhance the quality of our paper.